# Multi-Array Visible-Light Optical Generalized Spatial Multiplexing–Multiple Input Multiple-Output System with Pearson Coefficient-Based Antenna Selection

Li Zhao, Hao Wang, Junlin Chen * and Xiangyan Meng

School of Electronic Information Engineering, Xi'an Technological University, Xi'an 710021, China;
pilly_lily@126.com (L.Z.); 13572563010@163.com (H.W.); wulizu@126.com (X.M.)
* Correspondence: 13474478227@163.com

**Abstract:** To address the limitations of poor environmental adaptability, unsatisfactory bit-error performance, and high complexity of conventional antenna selection algorithms applied to a multi-array visible-light optical generalized spatial multiplexing–multiple-input multiple-output (OGSMP-MIMO) system, an OGSMP-MIMO system based on Pearson coefficient antenna selection is proposed. The algorithm adopts the correlation of Pearson coefficients among photoelectric detector (PD) terminals at different positions and active transmit antennas to select the optimal antenna combination without relying on the accuracy of channel estimation, for realizing the multiplexing of the time and space domains, and to improve the bit-error performance. Finally, experiments were conducted to verify the feasibility of the antenna selection algorithm, based on the Pearson coefficients. The results indicated that when the bit-error rate reached $10^{-6}$, using the antenna selection algorithm based on the Pearson coefficient, the signal-to-noise ratio was improved by 2.7 dB and 3.7 dB when compared with the norm-based antenna and random selection algorithms, respectively. In addition, increasing the number of active transmitting antennas can improve the transmission rate; however, the bit-error performance will be compromised. In the same modulation mode, increasing the number of transmitting antennas will reduce the bit-error performance.

**Keywords:** OGSMP-MIMO system; Pearson coefficient; piecewise bound theory; norm-based antenna selection





## 1. Introduction

The development of visible-light communication technology (VLC) [1–4] has always focused on improving the transmission rate and quality under harsh channel environments and limited bandwidth. It has been verified that the most effective way of improving the data transmission rate and quality on the visible-light channel is by adopting the multiple-input multiple-output (MIMO) [5–8] technology. In other words, multiple antennas or antenna arrays are placed at the sending and receiving ends of the visible-light communication system to transmit information. MIMO technology is a key technology widely used in the new generation of visible-light communication systems to double the capacity and spectral efficiency of communication systems without increasing their bandwidth. The MIMO technology has been adopted in indoor optical communication systems.

Yesilkaya et al. proposed a novel generalized light-emitting diode (LED) refractive index modulation method for VLC systems based on multiple-input multiple-output (MIMO) orthogonal frequency-division multiplexing (OFDM). By adopting spatial multiplexing and LED index modulation, this method circumvents the typical spectral efficiency loss triggered by time-domain and frequency-domain shaping in OFDM signals; however, the complexity is extremely high [9]. Therefore, Mesleh et al. proposed OSM technology based on MIMO technology, which reduced the complexity but did not significantly improve the transmission rate [10]. In addition to utilizing conventional modulation symbols to

transmit information, optical generalized spatial modulation (OSM) [11–14] also hides part of the information in the index of the transmitting antenna, thereby improving the MIMO technology. Vasavada et al. proposed three low-complexity detection schemes for SM MIMO systems. The MRC-based scheme can achieve near-optimal performance while reducing the complexity of the ML-based SM receiver; however, it does not significantly improve the transmission rate [15]. Accordingly, researchers have proposed an optical generalized spatial modulation (OGSM) algorithm [16–20]. However, the definition of GSM in the current literature is not accurate. Chen Chen re-define GSM as follows: for the one where the activated transmitters transmit the same signal, it is defined as "GSM"; for the one where the activated transmitters transmit different signals, it is defined as "GSMP" [21]. Wang et al. optimized the amplitude phase modulation (APM) symbol constellation in a VLC system based on the GSM, and adopted the statistical convergence gradient descent (SCGD) algorithm. Although the numerical simulation verified the superior performance of the optimal constellation, the algorithm was too complex [22]. Lin et al. proposed a generalized precoder design formula and iterative algorithm for the spatial modulation of MIMO systems with CSIT, to address the high complexity of the maximum-likelihood detection algorithm in large-scale GSM scenarios. Although this algorithm significantly reduces the complexity, it does not optimize the transmission performance [23]. Kumar et al. proposed a generalized spatial modulation (GSM) multiple-input, multiple-output coding scheme based on an active space and cooperative constellation. Compared with the conventional GSM algorithm, this algorithm improves the power efficiency of indoor visible-light communication; however, its complexity is high [24]. Lu et al. proposed a novel convolutional coding-optical generalized spatial modulation-space diversity (CC-OGSM-SD) serial relay system under the M distribution [25]. Based on the OGSM technology, it was proposed to improve the bit-error performance using an antenna selection algorithm. Robert proposes an antenna selection algorithm based on norm selection, which requires known channel parameters, but the actual channel is time-varying [26]; ZAPPONE et al. combined the fractional programming theory and the framework of continuous convex optimization problems, to propose a fractional programming power allocation scheme [27]. Du Liping et al. proposed an optimization method for antenna selection and power allocation, based on genetic algorithms [28]. The Pearson correlation coefficient is widely used to measure the degree of correlation between two variables, and its value is between −1 and 1. The larger the absolute value, the stronger the correlation [29]. In this paper, we propose to use the Pearson coefficient to express the correlation between the central luminous intensity of the LED and the illuminance received by the photodetector, and then to select the optimal antenna combination.

The aforementioned studies, despite their characteristics, exhibited a poor scope of application or high complexity. Therefore, this study proposes a multi-array visible-light OGSMP-MIMO system design scheme, based on Pearson coefficients. The basic principle of this scheme is to adopt the Pearson coefficient correlation between the photoelectric detector (PD) end at different positions and the active transmitting antenna, to select the optimal antenna combination, independent of the accuracy of channel estimation, realize multiplexing of the time and space domains, and improve the bit-error performance.

## 2. OGSMP-MIMO System Modeling

### 2.1. Channel Modeling

In the visible-light indoor lighting scenario, to satisfy the lighting and communication requirements simultaneously, multiple LEDs were arranged on the roof to establish a VLC communication link model, as illustrated in Figure 1.

In an indoor space with length L, width W, and height H, a three-dimensional coordinate system was established with a corner of the ground as the coordinate origin. *M* LEDs for lighting and communication were arranged on the roof, *N* PDs were arranged at the target end to be measured, and the communication between the target to be measured and each LED light source followed the VLC communication link model.

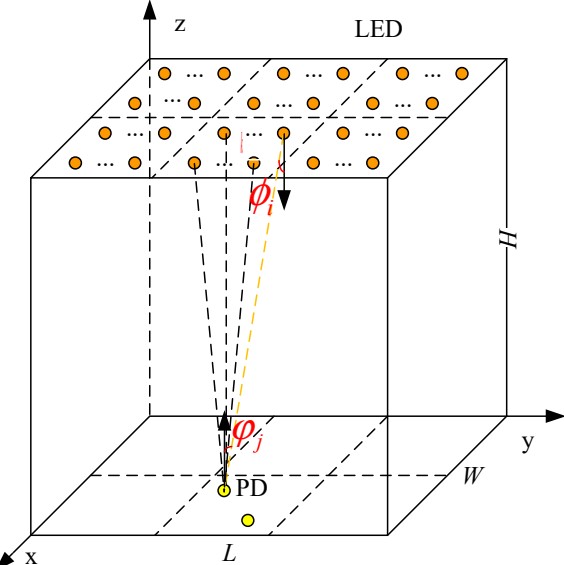

**Figure 1.** VLC communication link model.

The LED transmitter, wireless channel, and photodetector receiver form a multi-array visible-light MIMO communication system. Because a single light source layout causes uneven illumination, it is necessary to utilize a multi-LED array scattered layout in large scenes, which leads to obvious differences in the illumination intensity of each LED received by the PD receiver; multipath interference problems exist, which may affect the communication quality. Consequently, the VLC communication link model was divided into $K$ small positioning areas, and only the LED light sources in this area were considered at the PD receiving end of the $k$-th area. In the $k$-th region, the MIMO communication system channel can be expressed as follows:

$$H = \begin{pmatrix} h_{1,1} & h_{1,2} & \cdots & h_{1,N_t} \\ h_{2,1} & h_{i,j} & \cdots & h_{2,N_t} \\ \cdots & \cdots & \cdots & \cdots \\ h_{N_r,1} & h_{N_r,2} & \cdots & h_{N_r,N_t} \end{pmatrix} \tag{1}$$

where $h_{N_r,N_t}$ represents the illuminance generated by the $N_t$-th transmitting antenna and received by the $N_r$-th receiving antenna, and $h_{i,j}$ represents the direct current gain of each channel. Accordingly, $h_{i,j}$ is expressed as:

$$h_{i,j} = \left\{ \begin{array}{ll} \frac{A_{rx}^j}{d_{ij}^2} I(\phi_i) \cos(\varphi_j) & 0 \leq \varphi_j \leq \varphi_c \\ 0 & \varphi_j \geq \varphi_c \end{array} \right\} \tag{2}$$

where $A_{rx}^j$ denotes the receiving area of the $j$-th PD, $d_{ij}$ is the distance from the LED to the PD, $\varphi_i$ represents the light outgoing radiation angle between the LED and PD, $\varphi_j$ is the light incident angle, $\varphi_c$ denotes the field angle of the PD light receiving range, and $I(\varphi_i)$ represents the light intensity of the $i$-th LED along the $\varphi_i$ direction.

### 2.2. OGSMP-MIMO System Principle and Detection Algorithm

Assume an OGSMP-MIMO system with an $N_t$ transmit antenna and $N_r$ receiving antenna, as illustrated in Figure 2; in each symbol time, only $N_a$ ($1 < N_a < N_t$) antennas are activated, and there are $N = C_{N_t}^{N_a}$ possible combinations of activated antennas. In contrast, the OGSMP-MIMO system requires that the number of active antenna combinations must be an integer power of two; hence, the actual number of antenna combinations used is $N_c = 2^{\lfloor \log_2 N \rfloor}$, where $\lfloor \cdot \rfloor$ represents a round-down function.

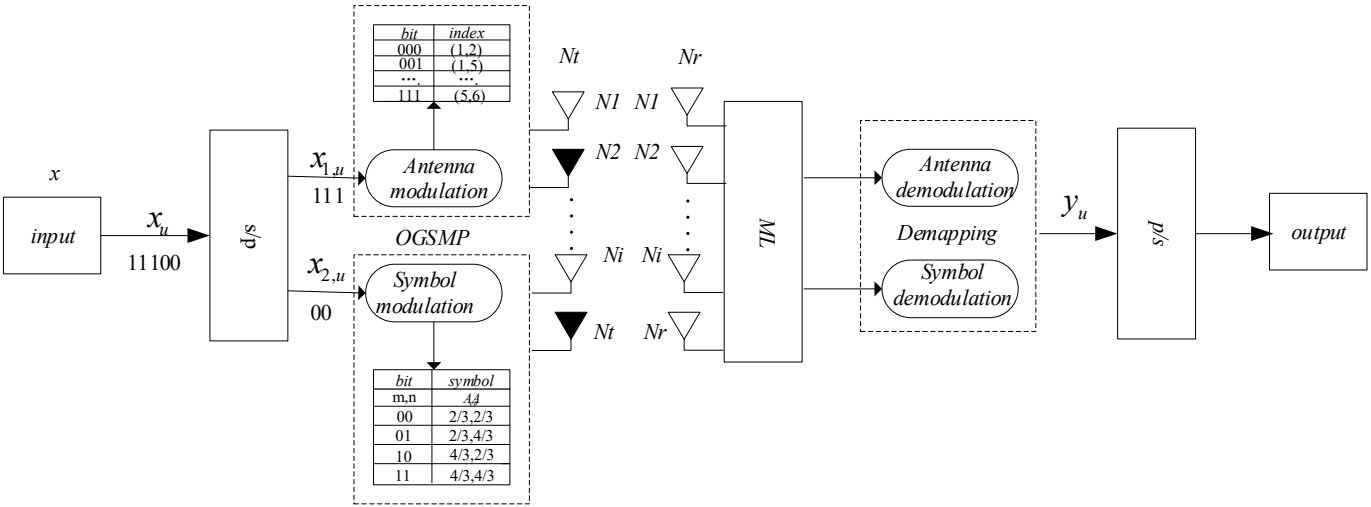

**Figure 2.** OGSMP-MIMO system model based on antenna selection.

The OGSMP-MIMO system sends independent modulation signals through an activated antenna combination. Assuming that the modulation order is *L*, the number of transmitted bits at the same time is as follows:

$$R_{OGSMP} = \left\lfloor \log_2 C_{N_t}^{N_a} \right\rfloor + N_a \log_2 L \tag{3}$$

As illustrated in Figure 2, the transmission bit of the OGSMP-MIMO system comprises two parts: the modulated signal and bit signal mapped by the antenna combination. The modulated bit stream is:

$$\boldsymbol{x_u} = [0, \cdots, 0, A_u, 0, \cdots, 0]^T \tag{4}$$

where $A_u$ denotes the modulation symbol transmitted by the *u*-th antenna. The modulated symbol vector is transmitted through the $N_r \times N_t$ dimensional channel *H*, where $\eta$ is the photoelectric conversion efficiency, which is usually taken as 1. The received signal model can be expressed as:

$$\boldsymbol{y} = \eta H x_u + n \; y \in C_{N_r \times N_t} \tag{5}$$

where $n \in C_{N_r \times N_t}$ represents the additive white Gaussian noise. At $A_t$ the receiving end, the sequence-number combination of the activated transmit antenna and the modulation symbol is discriminated by the maximum likelihood detection algorithm, and the original bit information is obtained by demapping [30]. The idea of the maximum likelihood (ML) algorithm is to obtain $H_x$ by multiplying the activated antenna combination by the channel matrix *H*, then calculate the Euclidean distance between *y* and $H_x$, and determine the activated antenna serial number and constellation symbol, according to the minimum value of the Euclidean distance. The formula for the maximum likelihood detection algorithm is

$$\left( \hat{k}, \hat{s} \right) = \underset{k,s}{\operatorname{argmin}} \| y - \eta H x \|_F^2 \tag{6}$$

where $\| \bullet \|_F$ represents the F-norm, $\hat{k}$ represents the estimated activated transmit antenna index, $\hat{s}$ represents the estimated value of the transmitted modulation symbol, and $\eta$ denotes the photoelectric conversion efficiency. *k* and *s* represent the index of the activated transmit antenna and exact value of the transmitted modulation symbol, respectively.

## 3. Antenna Selection Algorithm Based on Pearson Coefficient

Antenna selection is a key issue in MIMO technology. By selecting one or more antennas with the best performance from among multiple-transmit and receiving antennas in a MIMO system, the system error performance can be improved. Presently, conventional

antenna selection algorithms mainly include random antenna selection algorithms and norm-based antenna selection algorithms [31]. The random selection algorithm is the lower limit for evaluating the performance of the antenna selection algorithm; the norm-based antenna selection algorithm must know the channel parameters; however, the actual channel is time-varying, and the transmitter cannot obtain the channel state. Accordingly, this paper proposed an antenna selection algorithm based on the Pearson coefficient. The basic idea is to adopt the Pearson coefficient correlation between the PD end and the active transmit antenna at different positions, to select the optimal antenna combination that does not depend on the accuracy of channel estimation and has better environmental adaptability.

As illustrated in Figure 1, first, the installation distance of each LED was determined according to the VLC communication link model, and then the division of the positioning area was determined according to the received signal strength (RSS) value of each LED received by the receiver. Several positioning points were selected in the positioning area, the received RSS value of each LED was collected, and the actual coordinates of the reference point at the positioning point were collected and stored in the fingerprint database. The $M$ LED is denoted as $LED_m$, $m = 1, 2, \cdots, M$, and can be expressed as:

$$LED_m = \{R_m, (x_m, y_m, z_m)|m = 1, 2, \cdots, M\} \tag{7}$$

where $R_m$ represents the central luminous intensity of the $m$-th LED and unit: lx; and $(x_m, y_m, z_m)$ denotes the coordinate of the $m$-th LED (m). There are $N$ PDs in the positioning area, denoted as $PDn$, $n = 1, 2 \cdots, N$, and the light intensity received from the $LED_m$ at $PDn$ can be denoted as $S_n^m$. Therefore, the fingerprint library can be defined as

$$f_n = \{\boldsymbol{S_n}, (x_n, y_n, z_n), t_n|n = 1, 2, \cdots, N\} \tag{8}$$

where $\boldsymbol{S_n} = \left[S_n^{(1)}, S_n^{(2)}, \cdots, S_n^{(m)}\right]$ represents the illuminance of each LED received at $PDn$, $(x_n, y_n, z_n)$ represent the coordinates of the PD (m), and $t_n$ represents the number of times that the illumination is collected by the PD.

To reduce the amount of data calculation during positioning and improve the positioning accuracy, the indoor positioning area is divided into $k$-positioning areas, denoted as $f_{n,k}$, $k \in \{1, 2, \cdots, K\}$, and the partition fingerprint library can be defined as the following:

$$f_{n,k} = \{S_{n,k}, (x_{n,k}, y_{n,k}, z_{n,k}), t_{n,k}|n = 1, 2, \cdots, N_k\} \tag{9}$$

In addition, the partition LED position can be obtained, which can be expressed as

$$LED_{m,k} = \{R_{m,k}, (x_{m,k}, y_{m,k}, z_{m,k})|m = 1, 2, \cdots, M_k\} \tag{10}$$

where $R_{m,k}$ represents the central luminous intensity of the $m$-th LED in the $k$-th area, unit: lx; and $(x_{n,k}, y_{n,k}, z_{n,k})$ represent the coordinates of the PD in the $k$-th area (m).

In the partition fingerprint library, the Pearson correlation coefficient is used to calculate the correlation between two points. The Pearson correlation coefficient is a linear correlation coefficient that is defined as the ratio of the product of the covariance and the standard deviation of the two points. The value of the Pearson's coefficient is between $-1$ and 1. The larger the value, the higher the correlation. Therefore, it can be adopted as a selection criterion for antenna combination. For the $k$-th partition, the correlation coefficient between any LED combination and the fingerprint database PD can be expressed as

$$\rho(LED_{m,k}, f_{n,k}) = \frac{\text{cov}(R_{m,k}, S_{n,k})}{\sigma R_{m,k} \cdot \sigma S_{n,k}} = \frac{\sum\limits_{m=1}^{M_k}\sum\limits_{n=1}^{N_k} R_{m,k} \cdot S_{n,k}^{(m)} - \sum\limits_{n=1}^{N_k} R_{m,k} \cdot \overline{S}_{n,k}}{\sqrt{\sum\limits_{m=1}^{M_k}\sum\limits_{n=1}^{N_k}(R_{m,k})^2 - R_{m,k}^2} \cdot \sqrt{\sum\limits_{m=1}^{M_k}\sum\limits_{n=1}^{N_k}\left(S_{n,k}^{(m)}\right)^2 - \sum\limits_{n=1}^{N_k}\left(\overline{S}_{n,k}\right)^2}} \tag{11}$$

where $\overline{S}_{n,k}$ denotes the average illuminance of the LED combination received by the $n$-th PD in the $k$-th partition fingerprint database, and $m$ LEDs with high similarity are used as the transmitting antenna combination.

Taking 4 m × 4 m × 3 m scene as the spatial model, the specific light source layout is shown in Figure 3. The coordinates of LED lights are the following: (0.5, 0.5, 3), (0.5, 1, 3), (0.5, 1.5, 3), (1, 0.5, 3), (1, 1, 3), (1, 1.5, 3), (2.5, 0.5, 3), (2.5, 1, 3), (2.5, 1.5, 3), (3, 0.5, 3), (3, 1, 3), (3, 1.5, 3), (0.5, 2.5, 3), (0.5, 3, 3), (0.5, 3.5, 3), (1, 2.5, 3), (1, 3, 3), (1, 3.5, 3), (2.5, 2.5, 3), (2.5, 3, 3), (2.5, 3.5, 3), (3, 2.5, 3), (3, 3, 3), (3, 3.5, 3). According to the VLC communication link model, it is assumed that there are six LEDs in the $k$-th partition fingerprint library, and their central luminous intensity is 21 (lx); then, according to Equation (10), the LED information in the first area can be expressed as:

$$LED_{m,k} = \left\{ \begin{array}{ccc} \{21, (0.5, 0.5, 3)\} & \{21, (0.5, 1, 3)\} & \{21, (0.5, 1.5, 3)\} \\ \{21, (1, 0.5, 3)\} & \{21, (1, 1, 3)\} & \{21, (1, 1.5, 3)\} \end{array} \right\} \tag{12}$$

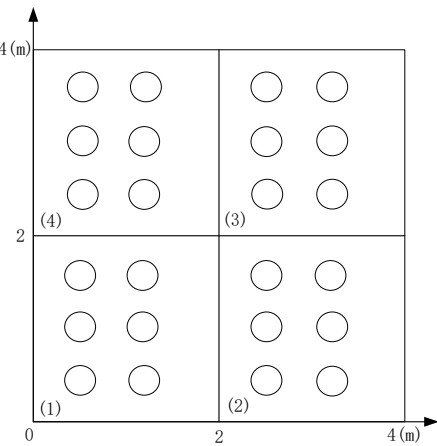

**Figure 3.** The layout of light sources in 4 m × 4 m × 3 m scene.

Using the visible-light communication model, the received illuminance of each PD can be obtained. Assuming that there are two PDs, and the coordinates are (0.8, 0.8, 0) and (0.8, 1.6, 0), there are 12 possibilities for receiving illuminance. According to Equation (9), the partition fingerprint database is expressed as

$$f_{n,k} = \left\{ \begin{array}{ccc} \left\{6.87_1^{(1)}, (0.8, 0.8, 0)\right\} & \left\{7.01_1^{(2)}, (0.8, 0.8, 0)\right\} & \left\{5.94_1^{(3)}, (0.8, 0.8, 0)\right\} \\ \left\{7.42_1^{(4)}, (0.8, 0.8, 0)\right\} & \left\{7.14_1^{(5)}, (0.8, 0.8, 0)\right\} & \left\{6.04_1^{(6)}, (0.8, 0.8, 0)\right\} \\ \left\{4.66_2^{(1)}, (0.8, 1.6, 0)\right\} & \left\{6.21_2^{(2)}, (0.8, 1.6, 0)\right\} & \left\{7.09_2^{(3)}, (0.8, 1.6, 0)\right\} \\ \left\{4.74_2^{(4)}, (0.8, 1.6, 0)\right\} & \left\{6.33_2^{(5)}, (0.8, 1.6, 0)\right\} & \left\{7.23_2^{(6)}, (0.8, 1.6, 0)\right\} \end{array} \right\} \tag{13}$$

where $\overline{S}_{n,k}$ represents the average illumination value at $t_n = 3$. Because six LEDs and two PD receivers are known, in the OGSMP-MIMO system, i.e., the transmitting antenna $N_t = 6$ and the receiving antenna $N_r = 2$, assuming that the transmitting antenna $N_a = 2$ is activated, there are a total of 15 antenna combinations in OGSMP-MIMO; however, the actual antenna combination of OGSMP-MIMO must satisfy the integer power of 2. Therefore, the actual antenna combination is selecting 8 of the 15 types. According to Equations (11)–(13), the set of Pearson coefficients can be obtained as

$$\rho = \left\{ \begin{array}{l} \rho_{1,2} = 0.9852, \rho_{1,3} = 0.9842, \rho_{1,4} = 0.9721, \rho_{1,5} = 0.9845, \rho_{1,6} = 0.9835 \\ \rho_{2,3} = 0.9962, \rho_{2,4} = 0.9831, \rho_{2,5} = 0.9975, \rho_{2,6} = 0.9961, \rho_{3,4} = 0.9818 \\ \rho_{3,5} = 0.9961, \rho_{3,6} = 0.9947, \rho_{4,5} = 0.9828, \rho_{4,6} = 0.9815, \rho_{5,6} = 0.9961 \end{array} \right\} \tag{14}$$

According to Equation (14), the actual antenna combination activated is

$$N_c = \{(1,2)\ (1,5)\ (2,3)\ (2,5)\ (2,6)\ (3,5)\ (3,6)\ (5,6)\} \tag{15}$$

Using the *L*-PAM modulation mode, the modulation order is *L* = 2, and the OGSMP-MIMO system mapping table is presented in Table 1.

**Table 1.** OGSMP-MIMO mapping based on antenna selection.

| Input Bit | Antenna Index Symbol | PAM Mapping Modulation Symbol | Output Signal after Mapping |
|---|---|---|---|
| 00000 | $[1\ 1\ 0\ 0\ 0\ 0]^T$ | $\frac{2}{3}, \frac{2}{3}$ | $\left[\frac{\sqrt{2}}{3}, \frac{\sqrt{2}}{3}, 0, 0, 0, 0\right]^T$ |
| 00001 | $[1\ 1\ 0\ 0\ 0\ 0]^T$ | $\frac{2}{3}, \frac{4}{3}$ | $\left[\frac{\sqrt{2}}{3}, \frac{2\sqrt{2}}{3}, 0, 0, 0, 0\right]^T$ |
| 00010 | $[1\ 1\ 0\ 0\ 0\ 0]^T$ | $\frac{4}{3}, \frac{2}{3}$ | $\left[\frac{2\sqrt{2}}{3}, \frac{\sqrt{2}}{3}, 0, 0, 0, 0\right]^T$ |
| 00011 | $[1\ 1\ 0\ 0\ 0\ 0]^T$ | $\frac{4}{3}, \frac{4}{3}$ | $\left[\frac{2\sqrt{2}}{3}, \frac{2\sqrt{2}}{3}, 0, 0, 0, 0\right]^T$ |
| ... | ... | ... | ... |
| 10000 | $[0\ 1\ 0\ 0\ 0\ 1]^T$ | $\frac{2}{3}, \frac{2}{3}$ | $\left[0, \frac{\sqrt{2}}{3}, 0, 0, 0, \frac{\sqrt{2}}{3}\right]^T$ |
| 10001 | $[0\ 1\ 0\ 0\ 0\ 1]^T$ | $\frac{2}{3}, \frac{4}{3}$ | $\left[0, \frac{2\sqrt{2}}{3}, 0, 0, 0, \frac{\sqrt{2}}{3}\right]^T$ |
| 10010 | $[0\ 1\ 0\ 0\ 0\ 1]^T$ | $\frac{4}{3}, \frac{2}{3}$ | $\left[0, \frac{\sqrt{2}}{3}, 0, 0, 0, \frac{2\sqrt{2}}{3}\right]^T$ |
| 10011 | $[0\ 1\ 0\ 0\ 0\ 1]^T$ | $\frac{4}{3}, \frac{4}{3}$ | $\left[0, \frac{2\sqrt{2}}{3}, 0, 0, 0, \frac{2\sqrt{2}}{3}\right]^T$ |
| ... | ... | ... | ... |
| 11100 | $[0\ 0\ 0\ 0\ 1\ 1]^T$ | $\frac{2}{3}, \frac{2}{3}$ | $\left[0, 0, 0, 0, \frac{\sqrt{2}}{3}, \frac{\sqrt{2}}{3}\right]^T$ |
| 11101 | $[0\ 0\ 0\ 0\ 1\ 1]^T$ | $\frac{2}{3}, \frac{4}{3}$ | $\left[0, 0, 0, 0, \frac{2\sqrt{2}}{3}, \frac{\sqrt{2}}{3}\right]^T$ |
| 11110 | $[0\ 0\ 0\ 0\ 1\ 1]^T$ | $\frac{4}{3}, \frac{2}{3}$ | $\left[0, 0, 0, 0, \frac{\sqrt{2}}{3}, \frac{2\sqrt{2}}{3}\right]^T$ |
| 11111 | $[0\ 0\ 0\ 0\ 1\ 1]^T$ | $\frac{4}{3}, \frac{4}{3}$ | $\left[0, 0, 0, 0, \frac{2\sqrt{2}}{3}, \frac{2\sqrt{2}}{3}\right]^T$ |

## 4. Bit-Error-Rate Analysis

Using the OGSMP-MIMO system with the number of transmit antennas $N_t = 6$ and the receiving antennas $N_r = 6$ as an example, *L*-PAM was adopted as the modulation method, and the BER performance of the OGSMP-MIMO system was analyzed and deduced, using segmental bound theory.

Assuming that the total error bits in the OGSMP-MIMO system are $m_e$, and the total data sent is $10^6$, then the system BER can be divided into index-position and constellation-symbol errors. The error bits caused by the index-position and constellation-symbol errors are denoted as $m_{e0}$ and $m_{e1}$, respectively. From Equation (4), the original bit stream signal can be determined; then, the received signal is restored as

$$\hat{x}_u = [0, \cdots, 0, \hat{A}_u, 0, \cdots, 0]^T \quad \hat{x}_u \in C_{10^6 \times 1} \tag{16}$$

Considering $P(x_u \to \hat{x}_u | H)$ to represent the conditional probability that the transmitted signal is $x_u$ and the receiving end recovers to $\hat{x}_u$ when the channel information $H$ is known, then using the union bound theory, the upper bound can be expressed as

$$m_e \leq \sum_u \sum_{\hat{u}} P(x_u \to \hat{x}_u | H) \frac{1}{10^6} \hat{m}_e(u, \hat{u}) \tag{17}$$

where $\hat{m}_e(u, \hat{u})$ represents the Hamming distance between $x_u$ and $\hat{x}_u$, and the expression form of the signal at the receiving end, according to $y_u = H_u x_u + n_u$, is

$$\boldsymbol{y_u} = [n_1, \cdots, H_u A_u + n_u, \cdots, n_{10^6}] \tag{18}$$

When the receiving end returns to $\hat{x}_u$, i.e., the antenna position index is incorrect, the conditions should be met at this time.

$$\|y_u - H_u x_u\|_F^2 > \|y_u - \hat{H}_u \hat{x}_u\|_F^2 \tag{19}$$

Equation (20) can be simplified to

$$2\text{Re}\left\{\left(\hat{H}_u\right)^H \hat{n}_u\right\} - 2\text{Re}\left\{(H_u)^H n_u\right\} > |H_u A_u|^2 + |\hat{H}_u \hat{A}_u|^2 \tag{20}$$

where $n_u$ represents the complex Gaussian noise of the $u$-th antenna, and defines $N = 2\text{Re}\left\{\left(\hat{H}_u\right)^H \hat{n}_u\right\} - 2\text{Re}\left\{(H_u)^H n_u\right\}$; accordingly, it can be deduced that it is a zero-mean Gaussian random variable:

$$N \sim N\left(0, 2|H_u A_u|^2 \cdot N_0 + 2|\hat{H}_u \hat{A}_u|^2 \cdot N_0\right) \tag{21}$$

Hence,

$$P(x_u \rightarrow \hat{x}_u | H) = Q\left(\sqrt{\frac{|H_u A_u|^2 + |\hat{H}_u \hat{A}_u|^2}{2N_0}}\right) \tag{22}$$

where $Q(x) = \int_x^\infty \exp\left(-t^2/2\right)/\sqrt{2\pi}dt$ and $P(x_u \rightarrow \hat{x}_u | H)$ denotes the conditional probability. In the actual simulation, the channel information generated in the simulation is substituted into Equation (23), and the final pairwise error probability $P(x_u \rightarrow \hat{x}_u | H)$ is obtained by averaging after multiple simulations. We denote $e(u, \hat{u})$ as the number of erroneous index bits when ascertaining the index position of the active antenna as $\hat{u}$, i.e., the Hamming distance between the two. Hence

$$m_{e0} = \frac{1}{10^6} \sum_{u=1}^{10^6} \sum_{\hat{u} \neq u=1}^{10^6} P(x_u \rightarrow \hat{x}_u) e(u, \hat{u}) \tag{23}$$

The constellation symbol error comprises two parts: $m_{e10}$ when the index position is incorrect, and $m_{e11}$, when the index position is correct. When the index position is wrong, the constellation symbol bits must be recovered by the data of the inactive antenna, leading to $\log_2 L/2$ error bits. Traverse all antenna index positions to obtain the error bit number of index position:

$$m_{e10} = \frac{1}{10^6} \sum_{u=1}^{10^6} \sum_{\hat{u} \neq u=1}^{10^6} P(x_u \rightarrow \hat{x}_u) \frac{\log_2 L}{2} \tag{24}$$

When the index position is correct, its probability upper bound position can be expressed as follows:

$$1 - \sum_{\hat{u} \neq u=1}^{10^6} P(x_u \rightarrow \hat{x}_u) \tag{25}$$

Let $P(y_u)$ denote the BER used to activate antenna detection when the index position is correct; i.e., the BER of $L$-PAM, and, for 2-PAM, it can be expressed as $P(y_u) = Q\left(\sqrt{2|H A_u|^2/N_0}\right)$; hence, we can obtain

$$m_{e11} = \frac{1}{10^6} \sum_{u=1}^{10^6} \left(1 - \prod_{\hat{u} \neq u=1}^{10^6} P(x_u \rightarrow \hat{x}_u)\right) \log_2 L P(y_u) \tag{26}$$

The total BER expression can be obtained from Equations (23), (24) and (26):

$$
\begin{aligned}
P_b = {} & \frac{1}{10^6} \sum_{u=1}^{10^6} \sum_{\hat{u} \neq u=1}^{10^6} P(x_u \to \hat{x}_u) e(u, \hat{u}) \\
& + \frac{1}{10^6} \sum_{u=1}^{10^6} \sum_{\hat{u} \neq u=1}^{10^6} P(x_u \to \hat{x}_u) \frac{\log_2 L}{2} \\
& + \frac{1}{10^6} \sum_{u=1}^{10^6} \left( 1 - \prod_{\hat{u} \neq u=1}^{10^6} P(x_u \to \hat{x}_u) \right) \log_2 L P(y_u)
\end{aligned}
\tag{27}
$$

## 5. Analysis of Simulation Results

### 5.1. Performance Analysis of OGSMP-MIMO System

Figure 4 shows the received signal eye pattern with different antenna selection algorithms under the condition of SNR(signal-to-noise ratio) of 10 dB. Figure 5 presents the bit-error rate of the OGSMP-MIMO system using different antenna selection algorithms. The system parameters are presented in Table 2.

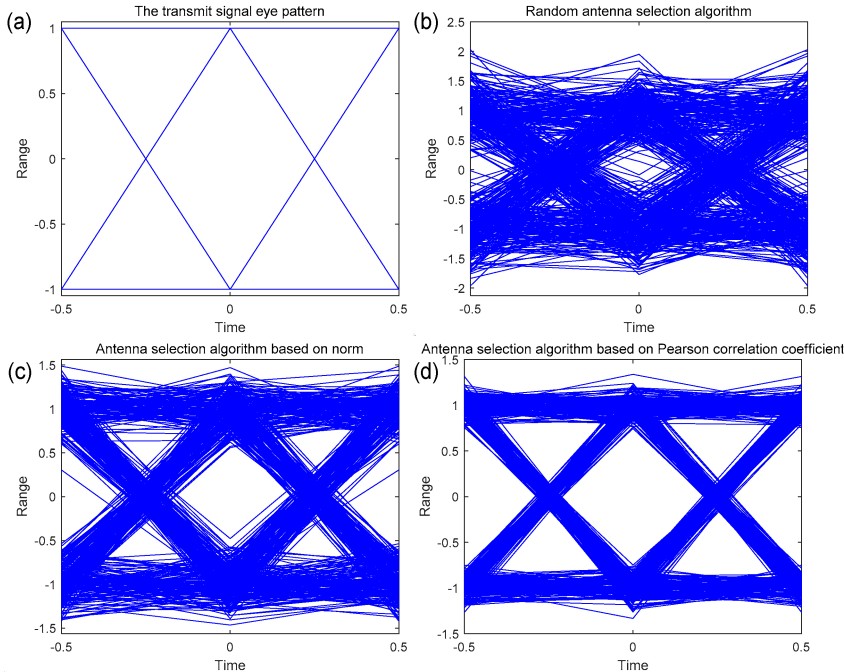

**Figure 4.** The received signal eye pattern with different antenna selection algorithms: ((**a**) is the transmit signal eye pattern, (**b**) is the received signal eye pattern based on the random antenna selection algorithm, (**c**) is the received signal eye pattern based on the norm antenna selection algorithm, and (**d**) is the received signal eye pattern based on the Pearson correlation coefficient antenna selection algorithm).

**Table 2.** Parameter information table.

| Number of Active Antennas $N_a$ | Number of Receiving Antennas $N_r$ | Modulation | Antenna Selection Algorithm |
|:---:|:---:|:---:|:---:|
| 2 | 2 | 2PAM | Random selection |
| 2 | 2 | 2PAM | Norm selection |
| 2 | 2 | 2PAM | Pearson selection |

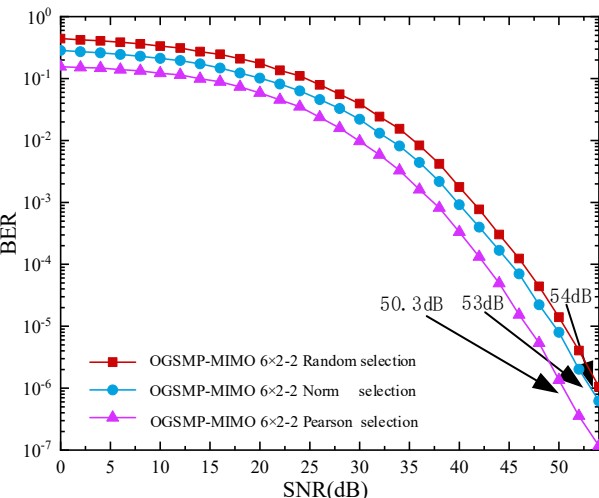

**Figure 5.** Bit-error rate of OGSMP-MIMO system with different antenna selection algorithms.

From the simulation results, it can be seen that the openness and shape of the received signal eye pattern based on the Pearson correlation coefficient antenna selection algorithm have been expanded and improved, compared to the random antenna selection algorithm and norm antenna selection algorithm, thereby ensuring maximum signal quality and minimum interference.

It can be deduced from the simulation results that, compared with the norm-based antenna selection algorithm and random selection algorithm, the OGSMP-MIMO system based on the Pearson coefficient antenna selection algorithm has better bit-error rate performance. When the bit-error rate reaches $10^{-6}$, the signal-to-noise ratio of the antenna selection algorithms based on the Pearson coefficient, antenna selection algorithms based on norm, and random selection algorithms reaches 50.3 dB, 53 dB, and 54 dB, respectively. The antenna selection algorithm based on the Pearson coefficient exhibits better bit-error performance, which is improved by 2.7 dB and 3.7 dB, respectively.

Figure 6 illustrates the theoretical and simulated bit-error performances of the OGSMP-MIMO system, based on the Pearson coefficients. The system parameters are the same as in Table 2.

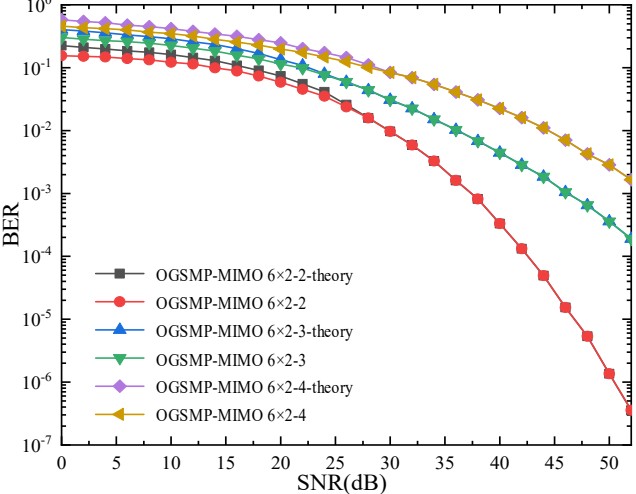

**Figure 6.** Theoretical and simulated bit-error performances of OGSMP-MIMO system based on Pearson coefficients.

It can be inferred from the simulation results that (1) when the SNR is relatively low, the theoretical BER of the OGSMP-MIMO system is higher than the actual BER, and when

the SNR is relatively high, the theoretical BER coincides with the actual BER, and (2) under the same modulation method, increasing the number of transmitting antennas will reduce the bit-error performance.

Figure 7 presents the bit-error rate of the OGSMP-MIMO system based on Pearson coefficients under different modulation orders. The system parameters are presented in Table 3.

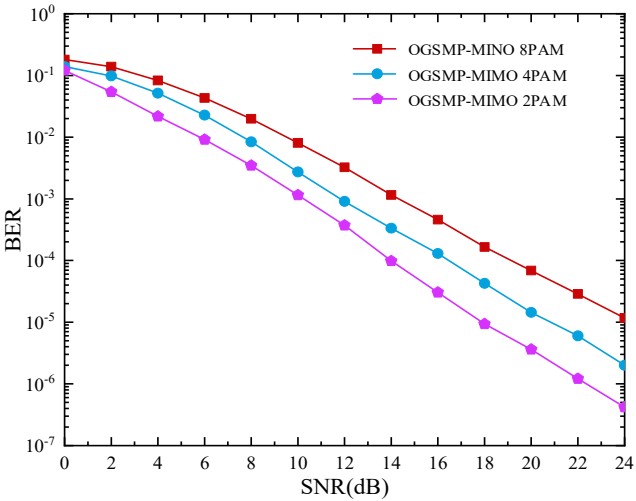

**Figure 7.** Bit-error rate of OGSMP-MIMO system based on Pearson coefficient under different modulation orders.

**Table 3.** Parameter information table.

| Number of Active Antennas $N_a$ | Number of Receiving Antennas $N_r$ | Modulation | Antenna Selection Algorithm |
|:---:|:---:|:---:|:---:|
| 2 | 6 | 2PAM | Pearson selection |
| 2 | 6 | 4PAM | Pearson selection |
| 2 | 6 | 8PAM | Pearson selection |

From the simulation results and Equation (3), it can be observed that, with an increase in the modulation order, the transmission rate increases; however, the bit-error performance decreases. The 2PAM, 4PAM, and 8PAM modulation modes were used to achieve transmission rates of 5 bpcu, 7 bpcu, and 9 bpcu, respectively (bpcu refers to the number of bits transmitted in each channel). The signal-to-noise ratios reached 10 dB, 12 dB, and 14 dB. Although the transmission rates were reduced by 2 bps and 4 bps, respectively, the bit-error performance was improved by approximately 2 dB and 4 dB, respectively.

*5.2. Experimental Verification*

The experimental platform illustrated in Figure 8 was built in a cube space with length, width, and height of 0.8 m, to further verify the feasibility of the antenna selection algorithm based on the Pearson coefficient in practical application scenarios. On the experimental platform, the space of the bottom surface 0.8 m × 0.8 m was divided into several areas at intervals of 0.1 m, and six LED light sources (RCW) with a power of 5 W were arranged on the top. Two PDs (LXD33MK) were arranged on the receiving plane, with the divided area as a reference positioning point, and the average value of the multiple illuminations of the light source in the PD was considered the RSS date.

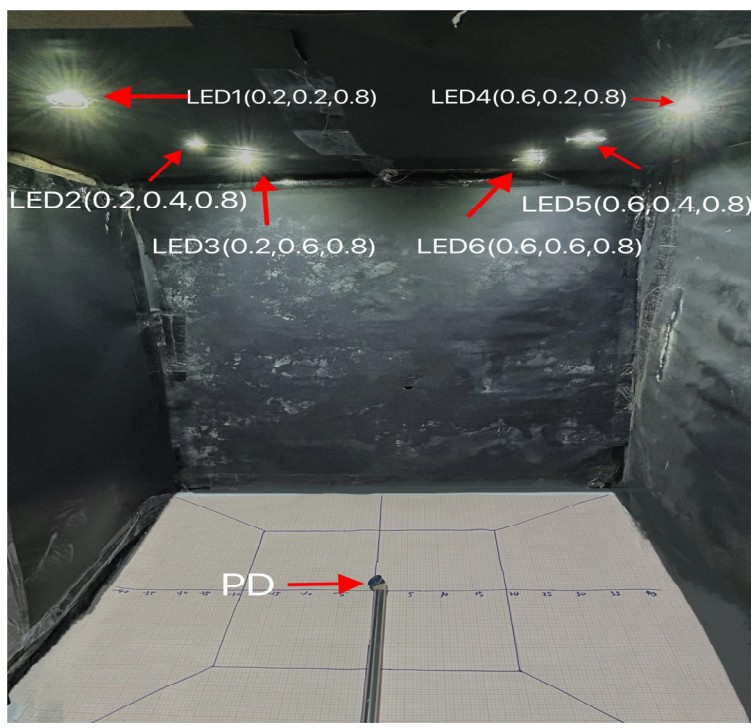

**Figure 8.** Actual experimental platform.

Assuming that there are two PD terminals, the coordinates are PD1 (0.8, 0.15, 0.35) and PD2 (0.8, 0.25, 0.35); there are four LEDs, and the coordinates are LED1 (0.2, 0.2, 0.8), LED2 (0.2, 0.4, 0.8), LED3 (0.2, 0.6, 0.8), LED4 (0.6, 0.2, 0.8), LED5 (0.6, 0.4, 0.8), and LED6 (0.6, 0.6, 0.8); the number of active antennas at the transmitting end $N_a = 2$, and the actual antenna combination of OGSMP-MIMO chooses eight types from fifteen. Combined with the actual experimental platform, it can be determined that there are eight types of illuminances received by different PDs from different LEDs, which can be expressed by Equation (9) as

$$
\begin{aligned}
P_b = {} & \frac{1}{10^6} \sum_{u=1}^{10^6} \sum_{\hat{u} \neq u=1}^{10^6} P(x_u \to \hat{x}_u) e(u, \hat{u}) \\
& + \frac{1}{10^6} \sum_{u=1}^{10^6} \sum_{\hat{u} \neq u=1}^{10^6} P(x_u \to \hat{x}_u) \frac{\log_2 L}{2} \\
& + \frac{1}{10^6} \sum_{u=1}^{10^6} \left(1 - \prod_{\hat{u} \neq u=1}^{10^6} P(x_u \to \hat{x}_u)\right) \log_2 L P(y_u)
\end{aligned}
\tag{28}
$$

According to Equations (11) and (28), the set of Pearson coefficients is

$$
\rho = \left\{
\begin{array}{l}
\rho_{1,2} = 0.9872, \rho_{1,3} = 0.8570, \rho_{1,4} = 0.7808, \rho_{1,5} = 0.8593, \rho_{1,6} = 0.7205 \\
\rho_{2,3} = 0.8170, \rho_{2,4} = 0.7542, \rho_{2,5} = 0.8177, \rho_{2,6} = 0.9459, \rho_{3,4} = 0.9395 \\
\rho_{3,5} = 0.9468, \rho_{3,6} = 0.8669, \rho_{4,5} = 0.9452, \rho_{4,6} = 0.8115, \rho_{5,6} = 0.9256
\end{array}
\right\}
\tag{29}
$$

According to Equation (30), the actual activated antenna combination is

$$
N_c = \{(1,2)\,(3,5)\,(2,6)\,(4,5)\,(3,5)\,(5,6)\,(3,6)\,(1,5)\}
\tag{30}
$$

*5.3. Spectral Efficiency, Transmission Rate and Complexity*

In addition to bit-error performance, complexity and spectral efficiency are important indicators for measuring system performance. In this study, an exhaustive number of ML

detection algorithms were adopted as the computational complexity, and the detection complexity of OGSMP-MIMO is

$$OGSM = N_c L^{N_a} \tag{31}$$

where $N_c$, $L$ and $N_a$ denote the actual transmit antenna combination, modulation order, and number of actually activated transmit antennas, respectively. According to Equation (3), the actual number of antenna combinations used is as follows:

$$m = \left\lfloor \log_2 C_{N_t}^{N_a} \right\rfloor \tag{32}$$

When $L$-PAM modulation is used, the number of modulation bits is

$$m_s = N_a \times \log_2(L) \tag{33}$$

Therefore, the complexity of OGSMP-MIMO is

$$OGSMP_{ML} = 2^{\left\lfloor \log_2 C_{N_t}^{N_a} \right\rfloor} \times 2^{N_a \times \log_2(L)} \times N_c L^{N_a} \tag{34}$$

Similarly, Table 4 can be obtained. It can be observed from Table 4 that the number of active antennas and the modulation order at the transmitter are the main factors that affect the system transmission rate, spectral efficiency, and complexity of the detection algorithm. When the system is determined, with an increase in the number of active antennas at the transmitter and in the modulation order, the transmission rate, spectral efficiency, and complexity significantly improve.

**Table 4.** Transmission rate and complexity of OGSMP-MIMO schemes with different modulation schemes.

| Modulation System | Transmission Rate/(bis/s) | Spectral Efficiency (bit/s Hz) | Complexity |
|---|---|---|---|
| OGSMP-2PAM | $\left\lfloor \log_2 C_{N_t}^{N_a} \right\rfloor + N_a$ | $\left\lfloor \log_2 \left( C_{N_t}^{N_a} \cdot 2^{N_a} \right) \right\rfloor$ | $2^{\left\lfloor \log_2 C_{N_t}^{N_a} \right\rfloor} \cdot 2^{N_a} \cdot N_c \cdot 2^{N_a}$ |
| OGSMP-4PAM | $\left\lfloor \log_2 C_{N_t}^{N_a} \right\rfloor + 2N_a$ | $2 \cdot \left\lfloor \log_2 \left( C_{N_t}^{N_a} \cdot 4^{N_a} \right) \right\rfloor$ | $2^{\left\lfloor \log_2 C_{N_t}^{N_a} \right\rfloor} \cdot 4^{N_a} \cdot N_c \cdot 4^{N_a}$ |
| OGSMP-8PAM | $\left\lfloor \log_2 C_{N_t}^{N_a} \right\rfloor + 3N_a$ | $3 \cdot \left\lfloor \log_2 \left( C_{N_t}^{N_a} \cdot 8^{N_a} \right) \right\rfloor$ | $2^{\left\lfloor \log_2 C_{N_t}^{N_a} \right\rfloor} \cdot 8^{N_a} \cdot N_c \cdot 8^{N_a}$ |

## 6. Conclusions

VLC systems usually adopt a multi-LED layout to take into account the dual functions of lighting and communication. Therefore, MIMO technology is required for multi-antenna cooperative transmission to realize high-speed communication. However, the MIMO system increases the channel capacity by simultaneously activating all the transmitting antennas, which leads to defects such as co-channel interference and multipath effects between channels, which limit the application of MIMO technology. Based on this, an antenna selection algorithm is introduced to improve the bit-error performance by selecting one or more antennas with the best performance from multiple transmitting and receiving antennas in the MIMO system. At present, the antenna selection algorithm mainly includes the random antenna selection algorithm, the norm antenna selection algorithm, and so on. The random selection algorithm randomly selects a set of antennas for data transmission without any selection criteria. The norm-based antenna selection algorithm must know the channel parameters; however, the actual channel is time-varying. Therefore, this paper proposes a Pearson coefficient antenna selection algorithm that does not depend on the accuracy of channel estimation. The basic principle is to use the Pearson coefficient correlation between PD terminals at different locations and active transmitting antennas to

select the optimal antenna combination, which further improves the bit-error performance of the OGSMP-MIMO system.

**Author Contributions:** Conceptualization, L.Z. and H.W.; methodology, L.Z.; software, J.C.; validation, L.Z., X.M. and J.C.; formal analysis, H.W.; investigation, J.C.; resources, L.Z.; data curation, H.W.; writing—original draft preparation, H.W.; writing—review and editing, L.Z.; visualization, J.C.; supervision, L.Z.; project administration, X.M.; funding acquisition, L.Z. All authors have read and agreed to the published version of the manuscript.

**Funding:** The study was supported by the National Defense Foundation of China (Grant No. 61671362), General project of industrial field of Shaanxi science and Technology Department (Grant No. 2022GY-072), Xi'an Science and technology project (Grant No. 2020KJRC0040).

**Institutional Review Board Statement:** Not applicable.

**Informed Consent Statement:** Not applicable.

**Data Availability Statement:** Data are contained within the article.

**Conflicts of Interest:** The authors declare no conflict of interest.

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
