# Peer review of "Multi-Array Visible-Light Optical Generalized Spatial Multiplexing–Multiple Input Multiple-Output System with Pearson Coefficient-Based Antenna Selection"

_photonics, doi:10.3390/photonics11010067_

Round 1
Reviewer 1 Report
Comments and Suggestions for Authors
This paper is devoted to the algorithm that adopts the correlation of Pearson coefficients between photoelectric detector terminals at different positions and active transmit antennas to select the optimal antenna combination without relying on the system channel characteristics. In my option, the paper could be published.
1、Increasing the number of active transmission antennas can improve transmission rate, but it will lose bit error performance. Whether there is an optimal point between transmission rate and bit error performance should be discussed.
2、The parameter ηis mentioned in formula 5, but the specific value of this parameter is not mentioned in the simulation model.
Comments on the Quality of English LanguageEnglish is acceptable, but there are places that can be improved.
Author Response
|
Comments 1: Increasing the number of active transmission antennas can improve transmission rate, but it will lose bit error performance. Whether there is an optimal point between transmission rate and bit error performance should be discussed. |
|
Response 1: According to Table 4, as the number of active antennas at the transmitting increases, the transmission rate and complexity will increase. An increase in complexity can lead to a decrease in system fault tolerance, resulting in a decrease in error rate performance. Therefore, based on Figure 6, it can be seen that selecting 6 transmitting and receiving antennas, 2 active antennas, and a modulation method of 4PAM can ensure error rate performance while improving transmission rate.
|
|
Comments 2: The parameter η is mentioned in formula 5, but the specific value of this parameter is not mentioned in the simulation model. |
|
Response 2: η is the photoelectric conversion efficiency. During the simulation experiment stage, setting η for 1. The description of this value has been provided on line 142.
|

Reviewer 2 Report
Comments and Suggestions for Authors
This article proposes an antenna selection algorithm based on Pearson coefficient to address the shortcomings of traditional antenna selection algorithms. It can improve the bit error performance of the OGSM-MIMO system. The theoretical analysis of the article is detailed and the experimental results are reliable. I suggest that this article be published in this journal.
1、In the article, the Pearson coefficient is used to express the correlation between the luminous intensity of the LED and the illumination received by the photo detector. Do you need to consider the reflecting light when selecting the optimal antenna combination? Will reflecting light have an impact on the optimal antenna combination?
2、Figure 4 represents theoretical and simulated-bit-error performances of OGSM-MIMO system based on Pearson coefficients,What is the difference between the two?
3、The article analyzes the bit error rate performance of different antenna selection algorithms. Can you provide the received signal eye pattern with different antenna selection algorithms?
Comments on the Quality of English LanguageMinor editing of English language required
Author Response
|
Comments 1: In the article, the Pearson coefficient is used to express the correlation between the luminous intensity of the LED and the illumination received by the photo detector. Do you need to consider the reflecting light when selecting the optimal antenna combination? Will reflecting light have an impact on the optimal antenna combination? |
|
Response 1: Reflected light does not need to be considered. Because the antenna selection algorithm based on Pearson coefficient designed in this paper is suitable for large scene environmental conditions, and reflected light can be ignored.
|
|
Comments 2: Figure 4 represents theoretical and simulated-bit-error performances of OGSM-MIMO system based on Pearson coefficients,What is the difference between the two? |
|
Response 2: The theoretical simulation performances are calculated according to the probability of sending symbol being falsely detected as symbol , which is calculated by the theory of joint boundary technology. The specific theoretical analysis and calculation process are the contents of the bit error rate analysis in Section 4 of this paper. The actual simulated-bit-error performances are achieved by randomly generating 108 binary information under specific simulation environment conditions. This information is processed through signal modulation, antenna selection, and other signal processing processes shown in Figure 2. Finally, the error rate performance is verified through simulation.
Comments 3: The article analyzes the bit error rate performance of different antenna selection algorithms. Can you provide the received signal eye pattern with different antenna selection algorithms? Response 3: The author has added the received signal eye pattern with different antenna selection algorithms in the article. Fig4 the received signal eye pattern with different antenna selection algorithms The above figure shows the received signal eye pattern with different antenna selection algorithms under the condition of SNR(signal-to-noise ratio) of 10dB. (a) is the transmit signal eye pattern, (b) is the received signal eye pattern based on random antenna selection algorithm, (c) is the received signal eye pattern based on the norm antenna selection algorithm, and (d) is the received signal eye pattern based on the Pearson correlation coefficient antenna selection algorithm. From the simulation results, it can be seen that the openness and shape of the received signal eye pattern based on Pearson correlation coefficient antenna selection algorithm have been expanded and improved compared to random antenna selection algorithm and norm antenna selection algorithm, thereby ensuring maximum signal quality and minimum interference.
For detailed information, please refer to Word |

Reviewer 3 Report
Comments and Suggestions for Authors
The authors study the antenna selection issue in OGSM-based MIMO-VLC systems and propose a Pearson coefficient-based antenna selection scheme to improve the system performance. Overall, the topic is timely and interesting, and the proposed scheme is verified to outperform existing benchmark schemes. This reviewer has the following comments for the authors to further improve the quality of the paper.
1. The title of the paper needs to be revised, as "antenna selection" should be highlighted in the title. Hence, this reviewer would suggest the following title "Multi-array visible light OGSM-MIMO system with Pearson coefficient-based antenna selection".
2. In Fig. 2, the blocks in figures should head upwards or head to the left side.
3. As per Eq. (6), a joint maximum likelihood detector is adopted in this work, which has high computational complexity. Instead, low-complexity detectors such as the one reported in the following paper [a] can be considered.
[a] OFDM-based generalized optical MIMO, Journal of Lightwave Technology, 2021.
4. As above, a more clear definition about generalized optical MIMO, including OGSM and OGSMP, has been provided in [a]. The authors should clearly defined the considered OGSM in the paper to avoid any concept misunderstanding.
5. It is claimed in Section 3 that Pearson coefficient-based antenna selection does not depend on the channel parameters. However, it still requires the positions of PDs and active transmitters, which can also be seen as "channel parameters". Hence, a better justification of this claim is required.
6. In Section 5, it is not necessary to use a table to provide the parameters for each figure, as the elements of each line in the table are (nearly) the same, see Table 3. So, simply giving the parameters in the descriptions is enough.
7. It is good to have experimental verification, but the provided experimental results are very limited. It would be much better if more experimental transmission results such as BER curves can be presented.
Comments on the Quality of English Language
The English usage and writting need to be further improved.
Author Response
|
Comments 1: The title of the paper needs to be revised, as "antenna selection" should be highlighted in the title. Hence, this reviewer would suggest the following title "Multi-array visible light OGSM-MIMO system with Pearson coefficient-based antenna selection". |
|
Response 1: Thanks for the reviewer's suggestion. The title of the paper has been revised to: "Multi-array visible light OGSM-MIMO system with Pearson coefficient-based antenna selection"
|
|
Comments 2: In Fig. 2, the blocks in figures should head upwards or head to the left side. |
|
Response 2: I'm sorry, I didn't quite understand the reviewer's comments, so I haven't made any changes in the text for now.
Comments 3: As per Eq. (6), a joint maximum likelihood detector is adopted in this work, which has high computational complexity. Instead, low-complexity detectors such as the one reported in the following paper [a] can be considered. [a] OFDM-based generalized optical MIMO, Journal of Lightwave Technology, 2021. Response 3: Thank you for recommending the references. The author has conducted a detailed analysis of the algorithms mentioned in the references. In the later stage, we will conduct research on Algorithm 1 and Algorithm 2 related to ML detection in the references.
Comments 4: As above, a more clear definition about generalized optical MIMO, including OGSM and OGSMP, has been provided in [a]. The authors should clearly defined the considered OGSM in the paper to avoid any concept misunderstanding. Response 4: In this literature, the author propose four OFDMbased generalized optical MIMO (GO-MIMO) techniques for bandlimited IM/DD OWC systems, including OFDM-based FD-GSM, FD-GSMP, TD-GSM and TD-GSMP. For MIMO transmission, spatial modulation (SM) can be considered as a digitized MIMO scheme, where only a single transmitter is selected to transmit signal at each time instant, which might not be able to fully explore the potential of MIMO transmission for spectral efficiency enhancement of bandlimited OWC systems. Lately, the concept of generalized spatial modulation (GSM) has been proposed to enhance the performance of conventional SM. However, the definition of GSM in the current literature is not accurate. This literature re-define the GSM as follows: for the one where the activated transmitters transmit the same signal, defined as “GSM”; for the one where the activated transmitters transmit different signals, defined as“GSMP”. In this paper, it can be seen from lines 130-133 and formula 3 that GSMP is used, which means that the activated transmitters transmit different signals. I have made modifications to GSM in the text and provided explanations for GSMP on lines 54-57.
Comments 5: It is claimed in Section 3 that Pearson coefficient based antenna selection does not depend on the channel parameters. However, it still requires the positions of PDs and active transmitters, which can also be seen as "channel parameters". Hence, a better justification of this claim is required. Response 5: The objective function of the norm antenna selection algorithm is: The principle of the norm antenna selection algorithm is to estimate the channel matrix and feedback the channel information to the transmitter. The transmitter selects the antenna combination by calculating the maximum norm of each column of the channel matrix, so it has a strong dependence on the accuracy of channel estimation. The Pearson coefficient correlation selection algorithm proposed in this article does not require channel estimation at the receiving end, nor does it require feedback the channel estimation values to the transmitter, so its dependence on the accuracy of channel estimation is not strong. In summary, the algorithm proposed in this article does not depend on the accuracy of channel estimation, rather than on the channel parameters. The modifications have been made in the text.
Comments 6: In Section 5, it is not necessary to use a table to provide the parameters for each figure, as the elements of each line in the table are (nearly) the same, see Table 3. So, simply giving the parameters in the descriptions is enough. Response 6: The parameters in Table 2 and Table 3 are nearly the same except for the antenna selection algorithm. Therefore, Table 3 has been deleted from the text and briefly explained on line 335-336.
Comments 7: It is good to have experimental verification, but the provided experimental results are very limited. It would be much better if more experimental transmission results such as BER curves can be presented. Response 7: The experimental model in this article is based on the RSS (received signal strength) visible light positioning model which is established by our research group. The core objective is to verify the feasibility of the Pearson coefficient antenna selection algorithm based on RSS in a multi LED light source layout environment. Through reasonable arrangement of light sources and detectors in 0.8m*0.8m*0.8m indoor space, as shown in Figure8. After multiple measurements, the irradiance of different light sources in the reference area was obtained, and the corresponding Pearson coefficients were calculated according to different irradiance distributions, so that the antenna can be effectively selected. Thus, the feasibility of the Pearson coefficient antenna selection algorithm based on RSS in the optical generalized space modulation system can be verified, which provides support for further research on the performance of the communication system. For detailed information, please refer to Word. |

Reviewer 4 Report
Comments and Suggestions for Authors
The paper has good technical content but must be improved as, in the reviewer’s opinion, there are some significant flaws. The written English of the document may also be improved.
The authors describe a VLC MIMO system where the selection of the transmitters (that the authors called antennas) is based on the Pearson coefficient correlation between receiver terminals at different locations and active transmitters to select the optimal antenna combination. It is stated several times in the paper that this approach does not require knowledge of the channel parameters, in contrast, for example, with a norm-based selection algorithm.
But the Pearson correlation, which the authors present in equation (11), depends on S,nk (the illuminance from source m received by receiver n, which in turn depends on the channel conditions defined by the matrix H. Hence, the reviewer argues that it is incorrect to state that the proposed method does not need to know the channel parameters since the amount of power received at each PD depends on the channel conditions. Still it remains valid the claim that the Pearson correlation performs better than the norm based, but unchanging channel conditions. If the channel parameters change, it remains to demonstrate that the proposed method still outperforms the norm-based one.
Some additional comments follow:
- Antenna is usually a term used in RF communications, it is strange to describe optical sources and receivers as antennas, even if they possess some lenses to collimate or focus the beam. Authors should clarify what is meant by antennas in this context of VLC communications.
- The paper focus much on the comparison with the norm-based algorithm. But there many papers in the literature with many variations of the method. A better review of the literature could have been provided.
- Figure 1: the caption should explain what is meant to be described by the diagram.
- Subscripts and superscripts are small and difficult to read
- line 180, explain what is meant by t_n, the definition does not seem correct
- line 201, eq (12) does derive directly from eq. 10 and only illustrates a particular case of the received illuminance and PD position. It should be clearly stated that this is a particular case and the justification for the values given should be explained. Moreover, authors should explain how the particular values of the received illuminance in eq. (9) were obtained, which should include distance and radiance angle, as given by (2).
- Acronym RSS is not defined
- In the illustration theoretical case six LEDs and two PDs were considered, whereas in the experimental verification a different case is considered: four LEDs and two PDs. Why using a different configuration? In fact no experimental measurements is actually done, only a different configuration is used again as an illustrative case. It is just a setup of the LED and PD disposition. The reviewer considers that this is not a valid experimental validation. Actual tests shoud be carried out to support the theory to be call it experimental verification.
Comments on the Quality of English Language
English can be improved.
Gave a few suggestions to the authors.
Author Response
|
Comments 1: The authors describe a VLC MIMO system where the selection of the transmitters (that the authors called antennas) is based on the Pearson coefficient correlation between receiver terminals at different locations and active transmitters to select the optimal antenna combination. It is stated several times in the paper that this approach does not require knowledge of the channel parameters, in contrast, for example, with a norm-based selection algorithm. But the Pearson correlation, which the authors present in equation (11), depends on S,nk (the illuminance from source m received by receiver n, which in turn depends on the channel conditions defined by the matrix H. Hence, the reviewer argues that it is incorrect to state that the proposed method does not need to know the channel parameters since the amount of power received at each PD depends on the channel conditions. Still it remains valid the claim that the Pearson correlation performs better than the norm based, but unchanging channel conditions. If the channel parameters change, it remains to demonstrate that the proposed method still outperforms the normbased one. |
|
Response 1: The objective function of the norm-based antenna selection algorithm is: The principle of the norm antenna selection algorithm is to estimate the channel matrix and feedback the channel information to the transmitter. The transmitter selects the antenna combination by calculating the maximum norm of each column of the channel matrix, so it has a strong dependence on the accuracy of channel estimation. The Pearson coefficient correlation selection algorithm proposed in this article does not require channel estimation at the receiving end, nor does it require feedback the channel estimation values to the transmitter, so its dependence on the accuracy of channel estimation is not strong. In summary, the algorithm proposed in this article does not depend on the accuracy of channel estimation, rather than on the channel parameters. The modifications have been made in the text.
|
|
Comments 2: Antenna is usually a term used in RF communications, it is strange to describe optical sources and receivers as antennas, even if they possess some lenses to collimate or focus the beam. Authors should clarify what is meant by antennas in this context of VLC communications. |
|
Response 2: In typical OWC systems, light-emitting diodes (LEDs) or laser diodes (LDs) are usually used as transmitters and photo-diodes (PDs) are employed as receivers. The research object of this paper is indoor visible light communication(VLC). In this system, LED needs to simultaneously consider the functions of lighting and information transmission. Therefore, in this paper, LEDs is referred to as the transmitting antenna and PDs is referred to as the receiving antenna.
Comments 3: The paper focus much on the comparison with the norm-based algorithm. But there many papers in the literature with many variations of the method. A better review of the literature could have been provided. Response 3: We have added relevant literature.
Comments 4: The caption should explain what is meant to be described by the diagram. Response 4: Figure 1 describes the VLC communication link model. The spatial model in Figure 1 is detailed introduction in lines 99-103, and the parameters in Figure 1 are explained in lines 118-121.
Comments 5: Subscripts and superscripts are small and difficult to read. Response 5: The author has adjusted the size of the subscripts and superscripts in the text.
Comments 6: line 180, explain what is meant by t_n, the definition does not seem correct. Response 6: In order to improve the measurement accuracy of fingerprint database, each positioning point is measured multiple times and then averaged. Therefore, in the text, tn represents the number of times that the illumination collected by the PD. The modification has been made on line 190-191 of the text.
Comments 7: line 201, eq (12) does derive directly from eq. 10 and only illustrates a particular case of the received illuminance and PD position. It should be clearly stated that this is a particular case and the justification for the values given should be explained. Moreover, authors should explain how the particular values of the received illuminance in eq. (9) were obtained, which should include distance and radiance angle, as given by (2). Response 7: In visible light communication systems, the layout of light sources needs to meet both lighting and communication requirements. To meet this requirement, this article takes 4m * 4m * 3m scene as the spatial model, and the specific light source layout is shown in the following figure.
Fig The layout of light sources in 4m * 4m * 3m scene The coordinates of LED lights are :(0.5,0.5,3),(0.5,1,3),(0.5,1.5,3),(1,0.5,3),(1,1,3), (1,1.5,3),(2.5,0.5,3),(2.5,1,3)(2.5,1.5,3),(3,0.5,3),(3,1,3),(3,1.5,3),(0.5,2.5,3),(0.5,3,3), (0.5,3.5,3),(1,2.5,3),(1,3,3),(1,3.5,3),(2.5,2.5,3),(2.5,3,3),(2.5,3.5,3),(3,2.5,3),(3,3,3), (3,3.5,3). The indoor lighting distribution under this layout mode is shown in the following figure. From the figure, it can be seen that under this layout, the indoor lighting intensity is greater than 300lx, and the uniformity of illumination is 90.5%, indicating that the layout can simultaneously meet the requirements of lighting and communication.
Fig The lighting distribution According to the principle of Pearson coefficient antenna selection algorithm described in the third section of this article. Firstly, it is necessary to divide the layout into regions. As shown in the above figure, the space of 4m*4m*3m scene is divided into four regions, and the coordinates in formula 12 are the coordinates of each LED in the first region. Assuming that there are two PDs, and the coordinates are (0.8, 0.8, 0) and (0.8, 1.6, 0), there are 12 possibilities for receiving illuminance. According to Eq. (9), the partition fingerprint database is expressed as formula 13.
Comments 8: Acronym RSS is not defined. Response 8: RSS refers to Received Signal Strength, which has been marked on line 176 where it first appears in the paper.
Comments 9: In the illustration theoretical case six LEDs and two PDs were considered, whereas in the experimental verification a different case is considered: four LEDs and two PDs. Why using a different configuration? In fact no experimental measurements is actually done, only a different configuration is used again as an illustrative case. It is just a setup of the LED and PD disposition. The reviewer considers that this is not a valid experimental validation. Actual tests shoud be carried out to support the theory to be call it experimental verification. Response 9: The experimental model in this article is based on the RSS (received signal strength) visible light positioning model which is established by our research group. The core objective is to verify the feasibility of the Pearson coefficient antenna selection algorithm based on RSS in a multi LED light source layout environment. Through reasonable arrangement of light sources and detectors in 0.8m*0.8m*0.8m indoor space, as shown in Figure 8. After multiple measurements, the irradiance of different light sources in the reference area was obtained, and the corresponding Pearson coefficients were calculated according to different irradiance distributions, so that the antenna can be effectively selected. Thus, the feasibility of the Pearson coefficient antenna selection algorithm based on RSS in the optical generalized space modulation system can be verified, which provides support for further research on the performance of the communication system. In order to maintain consistency with the theoretical analysis, we conducted a new experimental. In the new experimental system, six LED light sources (RCW) with a power of 5W were arranged on the top,two PDs (LXD33MK) were arranged on the receiving plane, and the specific experimental results are shown in Section 5.2. 5.2 Experimental verification The experimental platform illustrated in Figure 6 was built in a cube space with length, width, and height of 0.8 m, to further verify the feasibility of the antenna selection algorithm based on the Pearson coefficient in practical application scenarios. On the experimental platform, the space of the bottom surface was divided into several areas at intervals of , and six LED light sources (RCW) with a power of 5 W were arranged on the top. Two PDs (LXD33MK) were arranged on the receiving plane, with the divided area as a reference positioning point, and the average value of the multiple illuminations of the light source in the PD was considered the RSS date. Figure 8. Actual experimental platform Assuming that there are two PD terminals, the coordinates are PD1 (0.8, 0.15, 0.35) and PD2 (0.8, 0.25, 0.35); there are four LEDs, and the coordinates are LED1(0.2, 0.2, 0.8), LED2(0.2, 0.4,0.8),LED3(0.2, 0.6,0.8), LED4(0.6,0.2,0.8), LED5(0.6,0.4,0.8), LED6(0.6,0.6,0.8), the number of active antennas at the transmitting end = 2, and the actual antenna combination of OGSM-MIMO is to choose eight types from fifteen. Combined with the actual experimental platform, it can be determined that there are eight types of illuminances received by different PDs from different LEDs, which can be expressed by Eq. (9) as (28) According to Eqs. (11) and (28), the set of Pearson coefficients is (29) According to Eq. (30), the actual activated antenna combination is (30) For detailed information, please refer to Word. |

Round 2
Reviewer 3 Report
Comments and Suggestions for Authors
The authors have satisfactorily addressed my concerns, and hence this reviewer would like to recommend this paper for publication.
Comments on the Quality of English LanguageThe Quality of English Language is fine.
Reviewer 4 Report
Comments and Suggestions for Authors
I wanted to express my appreciation for the commendable effort you've put into addressing the comments and revisions for your manuscript. Your thorough and satisfactory responses have significantly strengthened the overall quality of the work.
It is evident that you've diligently incorporated the suggested changes, contributing to the enhancement of the manuscript. However, upon a final review, I noticed some minor issues with the layout that may benefit from correction before the printing stage. These are minor details, such as continuation of a sentence following an equation that should not be indented, I believe may be corrected during the final proofreading.
I trust that your attention to these remaining details will further refine the manuscript, and I look forward to the final version.